# Unraveling Therapeutic Opportunities and the Diagnostic Potential of microRNAs for Human Lung Cancer

**DOI:** 10.3390/pharmaceutics15082061

**Published:** 2023-07-31

**Authors:** Osama Sweef, Elsayed Zaabout, Ahmed Bakheet, Mohamed Halawa, Ibrahim Gad, Mohamed Akela, Ehab Tousson, Ashraf Abdelghany, Saori Furuta

**Affiliations:** 1Division of Cancer Biology, Department of Medicine, MetroHealth Medical Center, Case Western Reserve University School of Medicine, Cleveland, OH 44109, USA; 2Department of Zoology, Faculty of Science, Tanta University, Tanta 31527, Egypt; 3Department of Neuroscience, Baylor College of Medicine, Houston, TX 77030, USA; 4Department of Pharmacology, University of Colorado Denver Anschutz Medical Campus, Aurora, CO 80045, USA; 5Department of Statistics and Mathematics, Faculty of Science, Tanta University, Tanta 31527, Egypt; 6Department of Biology, College of Science and Humanities in Al-Kharj, Prince Sattam bin Abdulaziz University, Al-Kharj 11942, Saudi Arabia; 7Biomedical Research Center of University of Granada, Excellence Research Unit “Modeling Nature” (MNat), University of Granada, 18016 Granada, Spain

**Keywords:** lung cancer, environmental carcinogens, microRNAs, delivery, therapeutics, diagnosis

## Abstract

Lung cancer is a major public health problem and a leading cause of cancer-related deaths worldwide. Despite advances in treatment options, the five-year survival rate for lung cancer patients remains low, emphasizing the urgent need for innovative diagnostic and therapeutic strategies. MicroRNAs (miRNAs) have emerged as potential biomarkers and therapeutic targets for lung cancer due to their crucial roles in regulating cell proliferation, differentiation, and apoptosis. For example, miR-34a and miR-150, once delivered to lung cancer via liposomes or nanoparticles, can inhibit tumor growth by downregulating critical cancer promoting genes. Conversely, miR-21 and miR-155, frequently overexpressed in lung cancer, are associated with increased cell proliferation, invasion, and chemotherapy resistance. In this review, we summarize the current knowledge of the roles of miRNAs in lung carcinogenesis, especially those induced by exposure to environmental pollutants, namely, arsenic and benzopyrene, which account for up to 1/10 of lung cancer cases. We then discuss the recent advances in miRNA-based cancer therapeutics and diagnostics. Such information will provide new insights into lung cancer pathogenesis and innovative diagnostic and therapeutic modalities based on miRNAs.

## 1. Introduction

Lung cancer is one of the leading causes of cancer-related deaths worldwide, affecting both developed and developing countries. Despite advancements in early detection and treatment, the prognosis of lung cancer remains poor, with a five-year survival rate of less than 20% [1]. The etiology of lung cancer is complex and multifactorial, contributed to by environmental, genetic, and lifestyle factors. Above all, exposure to environmental pollutants, such as arsenic and benzopyrene (BaP), accounts for up to 1/10 of lung cancer cases [2]. Arsenic is commonly found in groundwater and soil, whereas BaP is present in cigarette smoke, diesel exhaust, and other combustion products [3]. Although the mechanisms by which these carcinogens induce lung cancer are still not fully understood, recent studies have unveiled pivotal contributions by miRNAs. MiRNAs are a class of small non-coding RNAs that play important roles in the post-transcriptional regulation of gene expression in eukaryotic cells. They are typically 18–25 nucleotides long and are involved in gene silencing, translational repression, and mRNA degradation [4]. MiRNAs bind to the 3′ untranslated region (UTR) of target mRNAs and induce their degradation or translational inhibition [5]. This regulates the expression of genes involved in a variety of biological processes, such as cell differentiation, proliferation, and apoptosis [6]. Dysregulation of discrete sets of miRNAs is implicated in numerous diseases, including cancer, cardiovascular disease, and neurological disorders [7]. New studies have indicated that miRNAs have significant functions in the development of lung cancer, specifically cases triggered by arsenic and BaP exposure, and could serve as viable targets for therapeutic intervention [8]. A set of miRNAs that regulate cancer-related signals like cell growth and proliferation have been observed to have varying levels of expression in lung cells following exposure to these carcinogens, demonstrating their involvement in the formation of cancer in the lungs caused by these toxins [9,10]. In this review, we will summarize the roles of miRNAs in lung carcinogenesis, especially in cases induced by exposure to arsenic and BaP, and discuss their diagnostic and therapeutic potentials. This review will help advance our insight into the role of miRNAs in lung cancer and justify their utility in improving patient outcomes.

## 2. The Genesis and Amplification of Human Lung Cancer

### 2.1. The Pathogenesis of Lung Cancer

Lung cancer progresses through multiple stages. The first stage, known as initiation, involves the occurrence of genetic mutations in normal cells, rendering them more vulnerable to the progression of cancer. These initiated cells then accumulate further genetic and epigenetic changes and begin to proliferate, forming pre-cancerous lesions termed dysplasia [11]. Over time, these pre-cancerous lesions progress to invasive cancer and eventually spread to other parts of the body. Lung cancer is divided into two major types: non-small cell lung cancer (NSCLC) (85% of cases) and small cell lung cancer (SCLC) (15% of cases), depending on the type of affected cells [12]. Recent advances in genomic and molecular profiling technologies have provided new insights into the genetic and molecular profiles of these two types, contributing to the development of specific targeted therapies [13]. However, lung cancer development is complex and multifactorial under the influences of numerous factors, including exposure to environmental carcinogens and lifestyle factors. Above all, tobacco smoke is the major environmental/lifestyle factor for lung cancer [14]. In addition, prolonged exposure to high concentrations of radon and air pollution, as well as work-related substances and secondhand smoke, increases the risks of lung cancer [15]. Thus, reducing exposure to these carcinogens will undoubtedly help to prevent lung cancer development [16]. Furthermore, genetic factors, including a family history of lung cancer and specific mutations, as well as gender, age, race, and ethnicity, could also influence the predisposition of individuals to the disease (Figure 1A) [17]. For example, a large proportion of lung cancers carry mutations in the EGFR, ALK, and KRAS genes. However, the mutation patterns can change over time and under targeted therapies, allowing them to acquire resistance towards previously effective treatments [18]. Such difficulty could possibly be circumvented by earlier detection and prompt treatment of this formidable disease [19].

### 2.2. Arsenic- and BaP-Induced Human Lung Carcinogenesis

Environmental exposure to arsenic and BaP serves as the major contributor to lung carcinogenesis [20]. Although exposure to each carcinogen increases the risk of lung cancer, co-exposure induces synergistic effects [21].

Arsenic is a naturally occurring metalloid widely distributed in the environment in soil, rocks, and minerals, while also being present in some groundwater sources [22]. Chronic exposure to arsenic is linked to lung cancer, skin cancer, bladder cancer, and other diseases [23]. Arsenic exists in several forms, including a highly toxic inorganic form that causes cancer, skin lesions, and cardiovascular disease [24]. Organic forms of arsenic, on the other hand, are less toxic and are typically found in plants and animals. Arsenic has been used for a variety of purposes throughout history, including pesticides, herbicides, insecticides, and certain industrial processes [25]. However, due to its toxicity, the use of arsenic has been restricted or banned in many countries, including the US. The mechanism of arsenic-induced lung cancer involves the induction of oxidative stress, DNA damage, and genomic instability, leading to gene mutations [26].

BaP is a polycyclic aromatic hydrocarbon (PAH) and a potent environmental carcinogen [27]. BaP is generated from burning organic materials, for example, the combustion of fossil fuels, tobacco smoking, grilling, or charring meat [28]. BaP is one of the most potent carcinogens found in cigarette smoke [29]. Exposure to BaP is linked to lung, skin, and bladder cancer. In addition, BaP elicits other harmful effects on human health, including respiratory problems, such as asthma and chronic bronchitis, as well as cardiovascular diseases [30]. BaP could also exert harmful effects on the ecosystem, particularly reproductive problems, developmental abnormalities, and the reduced growth and survival of aquatic organisms [31]. BaP is particularly harmful because it can easily enter the body through inhalation, ingestion, or skin contact. Once inside the body, BaP is converted into highly reactive metabolites that damage DNA and other cellular components [32]. The mechanism of BaP-induced lung cancer involves the formation of DNA adducts, triggering mutations in critical tumor suppressor genes and oncogenes [33]. One of the genes affected by BaP exposure is p53, a critical tumor suppressor gene regulating cell cycle arrest, DNA repair, and apoptosis. Mutations in p53 are found in many types of cancer and are associated with poor prognoses [34]. BaP exposure also causes epigenetic changes, such as DNA methylation, which could silence the expression of tumor suppressor genes and promote cancer development. BaP exposure may also activate oncogenes such as c-Myc, which is a transcription factor that regulates cell proliferation and apoptosis [35]. BaP additionally promotes the expression of growth factors and their receptors, such as the epidermal growth factor (EGF) and EGF receptor (EGFR), the major contributors to lung cancer development [36].

As described above, arsenic and BaP induce lung cancer through different mechanisms. Arsenic interferes with DNA damage repair, increases oxidative stress, and promotes cell proliferation. BaP, on the other hand, causes mutations in tumor suppressor genes, disrupts cell signaling pathways, and suppresses the immune system [37]. When arsenic and BaP are present together, however, their harmful effects are amplified and synergized, serving as a profound risk factor for lung cancer [38]. For example, their co-existence leads to upregulation of the pro-tumor mitogen-activated protein kinase (MAPK) pathway involved in cell growth and survival and cancer development [39]. In addition, these carcinogens alter the expression of genes involved in DNA damage repair, cell cycle progression, and apoptosis, further contributing to carcinogenesis [40]. Moreover, co-exposure to both carcinogens could impair cellular defense mechanisms that otherwise protect lung cells from environmental toxins. For example, arsenic interferes with glutathione, a critical antioxidant that helps neutralize reactive oxygen species (ROS) and prevents oxidative damage [41]. BaP, on the other hand, antagonizes aryl hydrocarbon receptors (AhRs), which play a key role in detoxifying environmental pollutants. When these defense mechanisms are compromised, lung cells become more vulnerable to the harmful effects of arsenic and BaP, further increasing the risk of lung cancer. Any preemptive measures, such as reducing exposure to these carcinogens, especially in occupational settings and areas with high environmental contamination, are essential for reducing the risk of lung cancer [42].

## 3. MiRNA-Based Mechanisms of Lung Carcinogenesis

Dysregulation of miRNA expression has been implicated in various diseases, including cancer. MiRNAs have been shown to play key roles in mediating lung carcinogenesis in response to environmental carcinogens (Appendix A) [43]. Recent research has demonstrated the potential utility of miRNAs as diagnostic and prognostic biomarkers for lung cancer. Moreover, miRNAs have been explored as therapeutic targets for lung cancer treatment, with promising results from preclinical studies. The targeted delivery of miRNA analogs and anti-miRNA oligonucleotides to cancer cells has emerged as a highly promising avenue for therapeutic advancement.

### 3.1. MiRNA Biogenesis and Regulatory Roles in Human Lung Cancer

MiRNA biogenesis is a crucial mechanism for the post-transcriptional regulation of gene expression in cells. It involves a series of enzymatic steps that result in the processing of primary miRNA transcripts into mature miRNAs, which then bind to target mRNAs and regulate their expression. MiRNA biogenesis is a multi-step process that involves the transcription of DNA into a primary miRNA (pri-miRNA) molecule by RNA polymerase II, followed by the processing of the pri-miRNA in the nucleus by the Drosha enzyme and its cofactor DGCR8 to produce a precursor miRNA (pre-miRNA) molecule [44]. The pre-miRNA is then exported to the cytoplasm, where it is cleaved by the Dicer enzyme to form a miRNA duplex. The duplex is then loaded into the RNA-induced silencing complex (RISC), which includes Argonaute (AGO) proteins [45]. The miRNA strand serves as a guide for the RISC complex to bind with target mRNA molecules that possess matching sequences. The RISC complex binds with the target mRNA, and this interaction can potentially cause degradation of the mRNA or its translational inhibition, which ultimately lead to the silencing of the gene [46]. MiRNA biogenesis is tightly regulated by a complex network of molecular interactions involving multiple protein complexes and regulatory factors. In fact, aberrant expression of key miRNA biogenesis factors, such as Drosha, DGCR8, Dicer, and Exportin-5, is found in various types of lung cancer [47]. In particular, decreased expression of Dicer is often found in lung cancer [48].

Dysregulation of specific sets of miRNAs has indeed been associated with the development of lung cancer. The dysregulation of miRNAs involved in cell proliferation, apoptosis, and metastasis can contribute to the progression of lung cancer [49]. In lung cancer, certain miRNAs that have oncogenic properties, such as miR-21, miR-155, and miR-221/222, are found to be elevated [50]. These miRNAs promote cell proliferation and metastasis, and they can target tumor suppressor genes like PTEN and PDCD4, leading to their downregulation and loss of their tumor-suppressing functions [51]. Furthermore, oncogenic miRNAs like miR-21, miR-155, and miR-221/222 have also been associated with chemotherapy resistance in lung cancer. They can regulate drug transporters, apoptosis, and DNA repair pathways, thereby affecting the efficacy of chemotherapy treatment [52]. On the other hand, tumor suppressor miRNAs, such as miR-34, let-7, and members of the miR-200 family, are downregulated in lung cancer [53]. These miRNAs normally inhibit cell growth and metastasis, but their reduced expression levels in lung cancer can contribute to uncontrolled cell growth and metastatic spread. For example, miR-34a, which targets oncogenes like c-Met and Notch1, is downregulated in lung cancer [54]. Furthermore, several miRNAs, including the miR-200 family and miR-205, have been shown to regulate the epithelial–mesenchymal transition (EMT) of lung cancer [55]. The EMT is a process in which epithelial cells lose their polarity and cell–cell adhesion and gain mesenchymal properties, promoting invasion and metastasis [56]. Other miRNAs, including miR-126 and miR-210, have been implicated in regulating the angiogenesis of lung cancer (Figure 1B) [57]. Angiogenesis is the process of forming new blood vessels, crucial for tumor growth and metastasis [58]. These dysregulated miRNAs also interact with various signaling pathways that are frequently mutated in lung cancer, such as the EGFR and KRAS pathways, further highlighting their roles in the disease [59]. Identification of dysregulated miRNAs in lung cancer may provide new opportunities for the development of miRNA-based therapeutic strategies.

### 3.2. MiRNAs Mediate Lung Carcinogenesis by Arsenic and BaP Co-Exposure

Exposure of lung cells to carcinogens, like arsenic and BaP, could lead not only to mutations in oncogenes and tumor suppressor genes, but also to changes in miRNA expression that facilitate cancer development (Figure 2A and Appendix A). For instance, pro-tumor miR-21, which targets tumor suppressor genes such as PTEN, PDCD4, and RECKS, is elevated in arsenic-exposed lung cells [60]. Conversely, antitumor miR-200c is downregulated in arsenic-exposed lung cells to promote the EMT [61]. Furthermore, antitumor miR-31, targeting the JAK/STAT pathway involved in cell proliferation and survival, is downregulated in BaP-exposed lung cells [62,63]. We will describe more details of molecular pathways regulated by miRNAs below. Given the potential roles of miRNAs in carcinogen-induced lung cancer, miRNA-based therapies may hold promise as a novel approach to treating this disease [64,65]. For example, a group of miRNAs, miR-21, miR-155, miR-200c, miR-145, miR-34a, miR-31, and miR-126, are linked to lung carcinogenesis whether or not it is induced by arsenic and BaP exposure (Figure 2A,B). They all regulate the expression of lung cancer-associated genes KRAS, c-Myc, SOCS1, SATB2, PTEN, PDCD4, Bcl-2, TGFBR2, ZEB1, Cyclin D1, ZEB2, RECK, EGFL7, and KLF4 (Figure 2C,D). Thus, modulation of the miRNA–cancer driver gene axis may represent a potential therapeutic approach for lung cancer. In fact, miRNA mimics have been utilized to restore the expression of tumor suppressor miRNAs, while miRNA inhibitors have been used to target oncogenic miRNAs [66]. Additionally, miRNA-targeting nanoparticles or exosomes have been developed as more targeted and efficient miRNA delivery systems [67].

## 4. Deciphering the Molecular Signaling Pathways of MiRNAs in Lung Cancer

To develop miRNA-targeted therapeutics, it is essential to know which signaling pathways regulate miRNAs. In lung cancer, a group of oncogenes or tumor suppressor genes are dysregulated, leading to aberrant expression of the downstream miRNAs. The dysregulated genes and pathways upstream of miRNAs include the epidermal growth factor receptor (EGFR), KRAS, PI3K-Akt-mTOR, Wnt, Notch, Hedgehog, TGF-β, JAK/STAT, NF-κB, and Hippo pathways. These genes are commonly upregulated in lung cancer and promote cell proliferation, invasion, survival, and therapeutic resistance (Figure 3 and Appendix A). The genes/pathways are regulated by miRNAs including miR-21, miR-31, miR-34a, miR-155, and miR-221/222 [68,69], and they are involved in the apoptosis, cell proliferation, angiogenesis, and metastasis of cancer cells.

## 5. Different Types and Mechanisms of MiRNA-Based Therapies for Lung Cancer

MiRNA-based therapies utilize different approaches, including inhibiting oncogenic miRNAs, restoring tumor suppressor miRNAs, modulating the immune response, and sensitizing cancer cells to chemotherapy and radiation therapy. These strategies demonstrate the versatility of miRNA-based therapies in targeting cancer and hold promise for improving treatment outcomes.

### 5.1. Inhibition of Oncogenic MiRNAs

Oncogenic miRNAs have been implicated in the progression of tumorigenesis by suppressing the expression of tumor suppressor genes. Consequently, targeting oncogenic miRNAs has emerged as a promising therapeutic strategy for the treatment of lung cancer. MiRNA-based therapeutics, such as anti-miRNA oligonucleotides (AMOs), locked nucleic acids (LNA), and antisense oligonucleotides (ASOs), have been developed to inhibit the function of these oncogenic miRNAs. Another approach, known as miRNA sponges, has also been explored for sequestering miRNAs and inhibiting their activity. Notably, studies focusing on AMOs specifically designed to target miR-21, an oncogenic miRNA, have demonstrated their effectiveness in restraining lung cancer cell proliferation and inducing apoptosis [70].

### 5.2. Restoration of Tumor Suppressor MiRNAs

Tumor suppressor miRNAs play a pivotal role in impeding tumor growth and metastasis. In the context of lung cancer, a notable observation is the downregulation of tumor suppressor miRNAs. Hence, an enticing strategy for treating lung cancer involves the restoration of these miRNAs. By reinstating tumor suppressor miRNAs, it is anticipated that the inhibition of tumor growth and suppression of metastasis can be achieved [71]. MiRNA therapeutics, such as miRNA mimics, can restore the function of tumor suppressor miRNAs by binding to their target genes and inhibiting their expression. For example, a miR-34a mimic has been shown to inhibit lung cancer cell growth and induce apoptosis by targeting multiple oncogenic genes [71].

### 5.3. Modulation of Immune Response

MiRNAs have been implicated in the regulation of immune cell functions, suggesting their potential involvement in the modulation of antitumor immune responses. Dysregulation of these miRNAs can lead to impaired immune responses against tumors. MiRNA-based therapeutics, including miRNA antagonists and mimics, offer a promising approach to modulate the expression of immune-related miRNAs. For instance, studies have demonstrated that the use of a miR-155 antagonist can enhance the antitumor immune response in lung cancer by increasing the expression of key immune mediators, such as interferon-γ and interleukin-2. This highlights the potential of targeting specific miRNAs to manipulate immune-related pathways and improve antitumor immunity [72].

### 5.4. Sensitization to Chemotherapy and Radiation Therapy

MiRNAs could also play roles in the regulation of resistance to chemotherapy and radiation therapy, and dysregulation of these miRNAs could confer therapeutic resistance. MiRNA therapeutics, such as miRNA inhibitors, can modulate the expression of drug-resistance-related miRNAs and sensitize lung cancer cells to cancer treatments. For example, a miR-221 inhibitor has been shown to target multiple drug-resistance-related genes and sensitize lung cancer cells to chemotherapy [73].

## 6. MiRNA Therapeutics and Delivery Methods

Over the past decade, an extensive array of therapeutics based on miRNAs has been meticulously crafted and extensively explored in preclinical settings. MiRNA-based treatments have demonstrated compelling efficacy in animal models, effectively restricting metastasis and offering promising prospects for combating cancer spread (Figure 4A,B). The development of effective delivery systems is a critical aspect of miRNA therapeutics. Various delivery methods have been developed and tested, including lipid-based delivery, viral vectors, exosomes, aptamers, peptide-based delivery, and electroporation (Figure 5). Lipid-based nanoparticles have been shown to be effective in delivering miRNA therapeutics to target cells and can be designed to selectively target specific tissues and organs. The use of nanocarriers for delivering miRNA therapeutics offers a potential solution to address off-target effects and toxicity concerns. By encapsulating miRNAs in nanocarriers, targeted delivery to lung cancer cells can be achieved. Through the incorporation of targeting ligands on the nanocarrier surface, specific binding to lung cancer cell receptors can be achieved, minimizing exposure to normal tissues and reducing off-target effects. Additionally, nanocarriers provide protection for miRNAs, improving their stability and bioavailability. Controlled release mechanisms ensure sustained and localized delivery to the tumor site. Nanocarriers can also be designed to possess other advantageous properties, such as enhanced cellular uptake and triggered release, further optimizing the therapeutic potential of miRNA-based treatments for lung cancer [74].

Viral vectors, such as adenoviruses and lentiviruses, can also be used as delivery vehicles for miRNA therapeutics [75]. Exosomes, small vesicles that are naturally produced by cells, have shown promise as delivery vehicles for miRNA therapeutics due to their ability to target specific cells and tissues [76]. Aptamers, small molecules that can specifically bind to target cells, have also been investigated for their potential as delivery vehicles for miRNA therapeutics [77]. Peptide-based delivery methods have been developed to target specific cell types or tissues and have been shown to be effective in delivering miRNA therapeutics to these targets [78]. Electroporation, which involves the use of an electric field to introduce miRNA molecules into cells, has also been explored as a method for delivering miRNA therapeutics to specific tissues [79]. While each delivery method has its own advantages and limitations, continued research in this area is necessary to optimize delivery systems and maximize the therapeutic potential of miRNA-based therapies. MiRNA therapeutics encompasses two distinct categories aimed at manipulating the expression of specific miRNAs: miRNA mimics and miRNA inhibitors. MiRNA mimics serve to enhance the expression of a particular miRNA, while miRNA inhibitors work to decrease its expression (Figure 4A).

### 6.1. MiRNA Mimics

MiRNA mimics are synthetic RNA molecules that mimic the function of endogenous miRNAs. They are designed to increase the expression of a specific miRNA that is downregulated in cancer cells, thereby restoring its tumor-suppressive function [80]. MiRNA mimics are typically chemically modified to enhance their stability and reduce off-target effects. One of the most widely used miRNA mimics is miR-34a, which is downregulated in lung cancer and functions as a tumor suppressor by regulating multiple oncogenic pathways [81]. Several preclinical studies have shown that systemic delivery of miR-34a mimics can inhibit lung tumor growth and metastasis in mouse models [82]. Another example of miRNA mimics is miR-16, which is downregulated in lung cancer and targets multiple oncogenes [83]. Delivery of miR-16 mimics has been shown to induce apoptosis and inhibit lung cancer cell proliferation [84].

### 6.2. MiRNA Inhibitors

MiRNA inhibitors, also known as antagomirs or anti-miRNAs, are synthetic RNA molecules that inhibit the function of endogenous miRNAs. They are designed to target and bind to the mature miRNA, thereby preventing its interaction with target mRNAs [85]. MiRNA inhibitors are also chemically modified to enhance their stability and reduce off-target effects. One of the most studied miRNA inhibitors is the miR-21 inhibitor, which targets a miRNA that is overexpressed in lung cancer and promotes tumor growth and metastasis [86]. In preclinical studies, systemic delivery of miR-21 inhibitors has been shown to inhibit lung tumor growth and sensitize cancer cells to chemotherapy [87,88]. Another example of miRNA inhibitors is the miR-155 inhibitor; miR-155 is upregulated in lung cancer and promotes tumor growth and immune evasion [89]. Delivery of miR-155 inhibitors has been shown to suppress lung tumor growth and enhance antitumor immune responses [90].

### 6.3. Delivery Methods for MiRNA Therapeutics

The success of miRNA therapeutics depends on their efficient delivery to the target tissues and cells. The delivery methods for miRNA therapeutics can be broadly classified into viral and non-viral vectors [91].

#### 6.3.1. Viral Vectors

Viral vectors are the most used delivery vehicles for miRNA therapeutics. They include retroviruses, lentiviruses, adenoviruses, and adeno-associated viruses (AAVs). These vectors are engineered to express the desired miRNA mimic or inhibitor and are capable of efficient transduction of both dividing and non-dividing cells (Figure 5). Several preclinical and clinical studies have shown the efficacy of viral-vector-based delivery of miRNA therapeutics for lung cancer treatment [92,93]. For example, a phase I clinical trial tested the safety and efficacy of intravenous delivery of a lentiviral vector expressing miR-16 in patients with advanced NSCLC. The results showed that the treatment was well-tolerated and resulted in stable disease in some patients [94].

#### 6.3.2. Non-Viral Vectors

Non-viral vectors for miRNA delivery are an attractive alternative to viral vectors because they are generally safer, less immunogenic, and more easily customizable. They include lipid-based nanoparticles, polymers, and inorganic nanoparticles. Non-viral vectors can be designed to encapsulate miRNA mimics or inhibitors and deliver them to the target cells through various mechanisms, such as endocytosis and membrane fusion [95,96]. Lipid-based nanoparticles are the most extensively studied non-viral vectors for miRNA delivery. They consist of a cationic lipid core and a polyethylene glycol (PEG) shell, which enhance their stability and reduce their immunogenicity [97]. Several preclinical studies have shown the efficacy of lipid-based nanoparticles in delivering miRNA therapeutics to lung cancer cells [98,99]. For example, a recent study demonstrated that the intravenous delivery of lipid-based nanoparticles containing miR-34a mimics can inhibit lung tumor growth and metastasis in a mouse model of NSCLC [100].

Polymers are another type of non-viral vector for miRNA delivery. They can be designed to have chemical and physical properties suitable for optimal stability, biocompatibility, and release kinetics. Polyethyleneimine (PEI) is one of the most used polymers for miRNA delivery because of its high cationic charge and ability to condense miRNAs into nanoparticles [101,102]. Several preclinical studies have shown the efficacy of PEI-based nanoparticles in delivering miRNA therapeutics to lung cancer cells [103,104]. Inorganic nanoparticles, such as gold nanoparticles and magnetic nanoparticles, are also being explored as non-viral vectors for miRNA delivery. They have unique physicochemical properties, such as high surface areas and magnetic responsiveness, making them applicable for magnetic-resonance-guided miRNA delivery to lung cancer cells [105,106].

## 7. Clinical Trials of MiRNA Therapeutics in Lung Cancer Treatment

There has been a surge in new lung cancer therapies utilizing miRNAs to alter the activity of lung cancer cells. Such interest stems from the effectiveness, reduced toxicity, and improved specificity of miRNA-based therapies compared to traditional cancer treatments. The safety and efficacy of each miRNA-based cancer therapy has been tested through clinical trials. Here, we summarize some of the recent clinical trials investigating the use of miRNA therapeutics in lung cancer treatment.

A phase I clinical trial evaluating the miR-34a mimic, MRX34, in patients with advanced solid tumors, including lung cancer, yielded compelling results. This groundbreaking study demonstrated the safety and efficacy of MRX34, as it was well tolerated by the patients. Encouragingly, a significant number of patients experienced positive outcomes, with two achieving a partial response and five stabilizing their diseases [107]. A Phase I/II clinical trial assessing MRX34’s safety and efficacy in patients with unresectable primary liver cancer or liver metastases, including cases originating from lung cancer, yielded promising results. The trial confirmed MRX34′s safety profile and provided evidence of its antitumor activity. Notably, three out of twenty-four patients achieved a partial response, while eight patients experienced disease stabilization. Another Phase I/IIa clinical trial evaluated the safety and efficacy of MRX34 in combination with the immune checkpoint inhibitor pembrolizumab in patients with advanced solid tumors, including NSCLC. The trial demonstrated that the combination therapy was safe and well-tolerated, with evidence of antitumor activity in some patients, including those who had previously progressed on immunotherapy. Specifically, out of seventeen patients, two patients achieved a partial response and six patients showed disease stabilization [108].

MiR-16 mimic TargomiRs was also tested in a phase I clinical trial for safety and pharmacokinetics in patients with advanced solid tumors, including lung cancer. The trial showed that TargomiRs was safe and well-tolerated, with no dose-limiting toxicities observed. In addition, TargomiRs demonstrated evidence of antitumor activity in some patients, with one patient achieving a partial response and four patients achieving disease stabilization [109]. Another Phase I/IIa clinical trial also evaluated the safety and efficacy of TargomiRs in combination with the chemotherapy drug docetaxel in patients with advanced NSCLC. The trial demonstrated that the combination therapy was well-tolerated and showed evidence of antitumor activity, with seven out of twelve patients achieving partial response or stable disease conditions [110]. These findings suggest that MRX34 and TargomiRs may hold promise as potential therapeutic options for lung cancer patients [111]. Combining miRNA therapeutics with immunotherapy or chemotherapy may enhance their antitumor effects.

## 8. Potential Benefits and Limitations of MiRNA Therapeutics in Lung Cancer Treatment

MiRNA therapeutics offers a potential new treatment modality for lung cancer, with several potential benefits over traditional therapies. However, there are also several limitations and challenges that must be overcome before these therapies can be widely adopted in clinics.

### 8.1. Potential Benefits of MiRNA Therapeutics in Lung Cancer Treatment

(I) Targeted Carriers: In miRNA therapeutics, lung cancer treatment involves the utilization of specific carriers to deliver miRNA molecules directly to cancer cells. These carriers, such as liposomes, nanoparticles, or viral vectors, are engineered to protect and transport therapeutic miRNAs to their intended targets within the cancerous tissue. By incorporating miRNAs into these carriers, their stability and bioavailability are enhanced, allowing for efficient delivery and cellular uptake. This targeted approach enables the miRNAs to selectively modulate the expression of cancer-associated genes, thereby exerting precise and potent antitumor effects while minimizing damage to healthy cells [112]. (II) Reduced Toxicity: The utilization of miRNA therapeutics presents a promising approach to mitigate toxicity and minimize side effects through a reduction in off-target effects. By specifically targeting the intended miRNAs, these therapeutic interventions can significantly minimize the likelihood of unintended impacts on other genes, thereby enhancing the safety profile of the treatment [113]. (III) Personalized Therapy: Personalized therapy holds great promise in the realm of miRNA-based treatments. The expression profiles of miRNAs exhibit significant variation among individuals, thereby highlighting the potential for targeted therapies aimed at specific miRNAs. By tailoring treatment strategies to address the unique miRNA landscape of each patient’s tumor, personalized therapies can be developed, offering more effective and precise interventions for improved patient outcomes. [114]. (IV) Combination Therapy: MiRNA therapeutics can be combined with other therapies, such as chemotherapy, radiation therapy, or immunotherapy, to enhance their antitumor effects. Such combinatorial therapy could potentially lead to improved outcomes in lung cancer patients [115]. (V) Overcoming Drug Resistance: MiRNA therapeutics hold significant potential in overcoming drug resistance, a major challenge in the treatment of lung cancer. By targeting dysregulated miRNAs, these interventions can modulate multiple genes and signaling pathways involved in resistance mechanisms. They can restore sensitivity to chemotherapy or targeted therapies by reversing the epithelial–mesenchymal transition, modifying drug efflux, and sensitizing resistant cells through the regulation of key genes and pathways. Additionally, miRNA-based therapies can be combined with existing treatments to enhance efficacy and counteract drug resistance by targeting cancer cells through multiple pathways [116].

### 8.2. Limitations and Challenges of MiRNA Therapeutics in Lung Cancer Treatment

(I) Delivery challenges: One of the major challenges for miRNA therapeutics is the difficulty of delivering miRNAs to tumor cells. MiRNA therapeutics are often delivered via nanoparticles or other delivery systems, which could be a little complex due to several reasons. Firstly, miRNAs are fragile molecules that can easily degrade in the harsh environment of the body. To protect them, specialized delivery systems such as nanoparticles are employed, which require careful design and optimization. Additionally, the delivery systems must be able to efficiently navigate through various biological barriers, such as the extracellular matrix, blood vessels, and cellular membranes, to reach the tumor cells [117]. (II) Off-Target Effects: Despite the specific design of miRNA therapeutics to target miRNAs, there remains a potential for these miRNAs to affect a diverse range of genes. This introduces the risk of off-target effects and toxicity in normal tissues. Off-target effects pose a concern in miRNA therapeutics as they have the potential to affect a broad range of genes, which may result in unintended consequences and toxicity in healthy tissues. However, the use of nanoparticles offers a promising strategy to mitigate these off-target effects. By encapsulating and delivering miRNA therapeutics within nanoparticles, their release can be tightly controlled, allowing for targeted delivery to specific cells or tissues of interest. This localized delivery approach reduces the likelihood of off-target effects in normal tissues, as the nanoparticles help to enhance the specificity and precision of miRNA therapeutics, maximizing their therapeutic potential while minimizing unintended impacts [118]. (III) Destruction by Immune Cells: Exogenous miRNAs face the risk of immune-cell-mediated elimination, as they have the potential to trigger an immune response, resulting in their destruction. This immune response can limit the efficacy and stability of exogenous miRNAs. Therefore, it is important to consider the immune response as a potential obstacle when utilizing exogenous miRNAs for therapeutic purposes [119]. (IV) Regulatory Challenges: MiRNA therapeutics is a relatively new class of therapeutics, and there remain regulatory hurdles to be overcome before they can be widely utilized in clinics. These challenges include issues related to manufacturing, quality control, and regulatory approval [120]. (V) Limited Clinical Data: Despite recent clinical trials of miRNA-based therapeutics for lung cancer, there is not enough evidence for their safety and effectiveness. To date, most clinical trials have focused on evaluating the safety and tolerability of miRNA-based therapies, and only a few have assessed their therapeutic efficacy. Thus, further clinical studies are awaited to validate the utility of miRNA-based therapies and their potential in lung cancer treatment [121].

## 9. Diagnostic Potential of MiRNA Signatures in Lung Carcinogenesis

In recent years, the convergence of artificial intelligence (AI) and miRNA therapeutics has shown promising potential in the field of lung cancer diagnosis and classification. AI algorithms have been developed to analyze miRNA expression patterns obtained from patient samples, enabling the identification of specific miRNA signatures associated with different subtypes or stages of lung cancer. By leveraging machine learning techniques, these algorithms can effectively classify lung cancer cases based on their miRNA profiles, providing valuable insights into disease prognosis and personalized treatment strategies [122]. The integration of AI with miRNA therapeutics offers a powerful approach for the precise targeting of dysregulated miRNAs, potentially leading to more effective and tailored treatments for lung cancer patients. Such advancements hold great promise for improving both the accuracy of diagnosis and the development of innovative therapeutic interventions [123].

Several miRNAs have been identified as displaying aberrant expression patterns that actively contribute to the advancement of lung cancer, thereby fostering malignancy [124]. Notably, miR-21, miR-155, and miR-34a frequently exhibit upregulated levels in lung cancer, while miR-126 and miR-145 manifest downregulation [125]. The dysregulation of these miRNAs in the context of lung cancer holds great potential for their utilization as biomarkers, imparting invaluable benefits in the realms of diagnosis, prognosis, and therapeutic monitoring [126]. MiRNA signatures may be used to differentiate lung cancer from non-cancerous lung lesions, such as chronic obstructive pulmonary disease (COPD) and pneumonia. For example, certain miRNA signatures (miR-17-5p, miR-21, miR-27a, and miR-222) could distinguish lung adenocarcinomas from healthy lung tissues with high sensitivity and specificity [127]. MiRNA signatures could also be used to predict the prognosis and therapeutic response of lung cancer patients. For example, a miRNA signature consisting of miR-221, miR-222, and miR-146a has been associated with poor survival in NSCLC patients [128]. Another study has identified a miRNA signature (miR-210, miR-192, and miR-21) that could predict the chemotherapy response of NSCLC patients [129].

### 9.1. Analysis of MiRNAs in Bodily Fluid for the Better Staging of Lung Cancer Progression

Timely detection and diagnosis of lung cancer play a vital role in enhancing survival rates and enabling optimal treatment outcomes. The assessment of miRNAs in bodily fluids like plasma, serum, and bronchoalveolar lavage fluid (BALF) is an emerging and promising approach for improved staging of lung cancer progression [130]. Aberrant expression of certain miRNAs has been identified in lung cancer, playing a role in its progression. Specifically, miR-21, miR-155, and miR-34a are often found to be upregulated, while miR-126 and miR-145 are frequently downregulated in lung cancer cases. Analyzing these miRNAs in bodily fluids could offer valuable insights into the staging of lung cancer and its progression, providing important information for clinical assessment [131].

Elevated levels of specific miRNAs, namely miR-210, miR-21, and miR-155, were found to be significantly higher in the serum of patients diagnosed with advanced lung cancer when compared to individuals with early-stage disease or healthy controls [132]. Similarly, increased levels of miR-155, miR-210, and miR-21 were observed in the plasma of patients with advanced-stage NSCLC compared to those with early-stage disease or healthy controls [133]. In the BALF of patients with metastatic lung cancer, the levels of miR-148a and miR-152 were found to be notably reduced compared to individuals with localized disease or healthy controls, indicating their potential as biomarkers for distinguishing between different stages of lung cancer [134]. Another study focused on identifying a miRNA signature associated with lymph node metastasis in NSCLC patients. The inclusion of miR-210, miR-21, miR-486-5p, and miR-375 in the signature holds promise as a predictive marker for lymph node involvement in NSCLC patients with lymph node metastasis [135]. These findings highlight the significance of miRNA levels in aiding clinicians with accurate diagnosis and effective management strategies for this specific patient population [136].

### 9.2. Potential of MiRNA Analyses of Bodily Fluid for Early Detection

Analysis of miRNAs in bodily fluid, such as serum, plasma, and BALF has emerged as a promising approach for the early detection of lung cancer. The levels of miR-205 and miR-21 were significantly higher in the serum of patients with early-stage lung cancer than in patients with benign lung nodules or healthy controls [137]. Similarly, the miRNA signature (miR-21, miR-210, and miR-155) in plasma distinguished between lung cancer patients and healthy controls with high accuracy, even at the earliest stages of the disease [138].

Analysis of miRNAs in bodily fluids has significant potential for the identification of subtype-specific biomarkers in lung cancer. Specifically, a unique miRNA signature comprising miR-29a, miR-let-7f, miR-23a, and miR-27a has been observed in BALF, demonstrating high accuracy in distinguishing between different subtypes of non-small cell lung cancer (NSCLC) [139]. Likewise, the presence of specific miRNAs in serum, such as miR-19a, miR-92a, and miR-29c, has been found to correlate with the EGFR mutation status in NSCLC patients [140]. Furthermore, miRNAs present in bodily fluids offer valuable insights into disease progression and treatment response. Notably, elevated levels of a miRNA signature consisting of miR-1225-5p, miR-328, and miR-548 in serum have been associated with tumor progression and survival in NSCLC patients [141]. Conversely, a decrease in plasma levels of miR-126 is linked to chemotherapy resistance in NSCLC patients [142].

## 10. Conclusions and Future Perspectives

In conclusion, miRNAs play crucial roles in the development and progression of lung cancer by regulating dysregulated signaling pathways and responding to environmental carcinogens such as arsenic and BaP. MiR-21 has been found to promote cell proliferation and angiogenesis in response to arsenic exposure, while miR-34a inhibits cell growth and induces apoptosis. Similarly, miR-21 and miR-31 are upregulated in response to BaP exposure, promoting cell proliferation, invasion, and the EMT. Certain miRNAs have demonstrated potential as targets for lung cancer treatment, such as miR-34a and miR-150, which have been delivered to lung cancer cells through liposomes and nanoparticles and have effectively suppressed tumor growth by reducing the activity of specific genes involved in cancer advancement. However, miR-21 and miR-155 are often overproduced in lung cancer and have been linked to heightened cell proliferation, invasion, and chemotherapy resistance. Further research is needed to better understand the roles of miRNAs in lung cancer and to develop more effective miRNA-based therapies. Additionally, improving the delivery and efficacy of miRNA-based therapies is crucial, while new strategies such as nanoparticle-based delivery systems and combination therapies are being explored. Furthermore, the use of extracellular miRNAs as biomarkers for lung cancer diagnosis and prognosis is a promising area of research and could be utilized to improve early detection and personalized treatment.

## Figures and Tables

**Figure 1 pharmaceutics-15-02061-f001:**
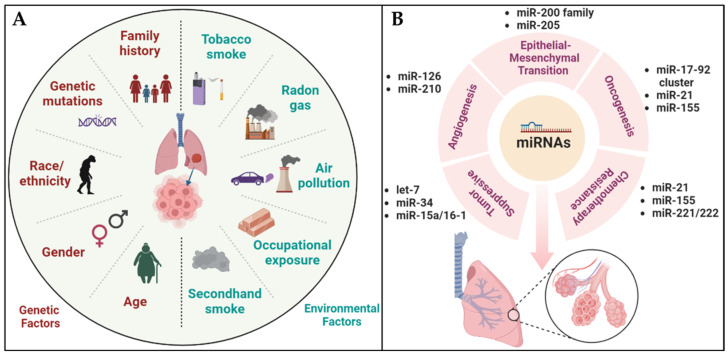
Representative diagrams of the causative factors of lung carcinogenesis and influence of miRNAs on lung cancer development. (**A**) A visual representation that shows how genetic and environmental factors come together to cause lung cancer. (**B**) A graphical portrayal of miRNAs with their different modulatory functions in lung cancer.

**Figure 2 pharmaceutics-15-02061-f002:**
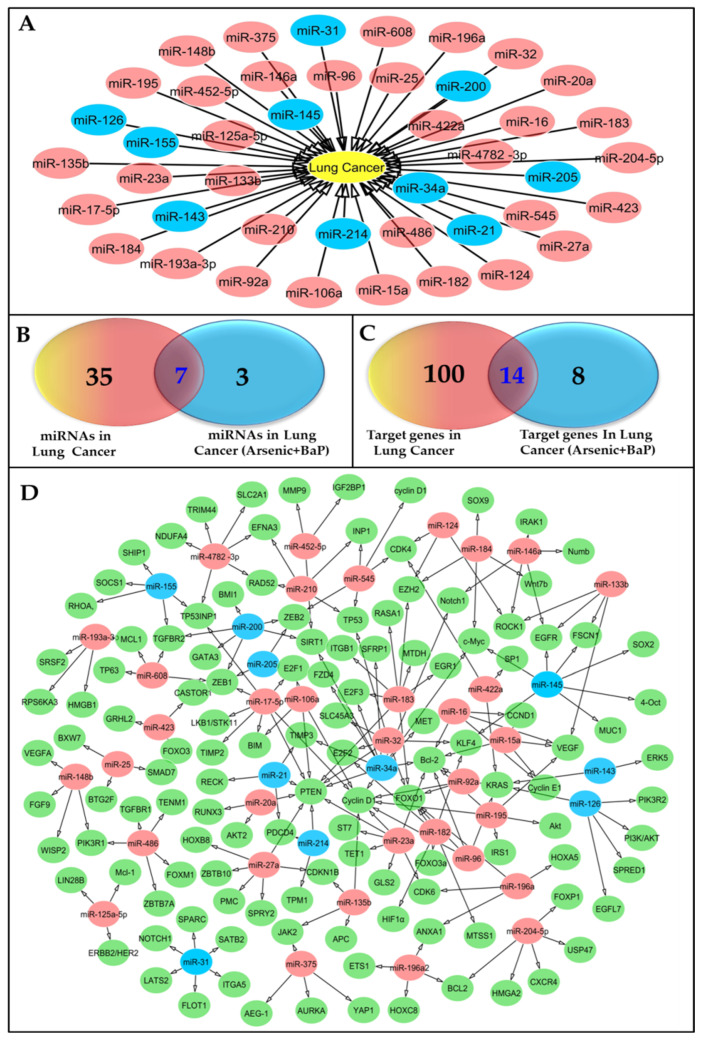
Representative diagrams of the miRNAs that have been associated with human lung cancer and their corresponding targets. (**A**) The implicated miRNAs in human lung cancer progression; those labeled in blue are involved in lung cancer induced by arsenic and BaP exposure. (**B**) The shared miRNAs that play a role in lung cancer and are also involved in the development of lung cancer caused by exposure to arsenic and BaP. (**C**) The target genes responsible for the progression of lung cancer and those that are also responsible for the development of lung cancer triggered by exposure to arsenic and BaP. (**D**) Representative diagram for miRNAs and their gene targets in lung cancer tissue. The miRNAs linked to lung cancer are depicted in red, whereas those associated with lung cancer induced by arsenic and BaP are indicated in blue. The genes that are targeted by these miRNAs are visually represented in green.

**Figure 3 pharmaceutics-15-02061-f003:**
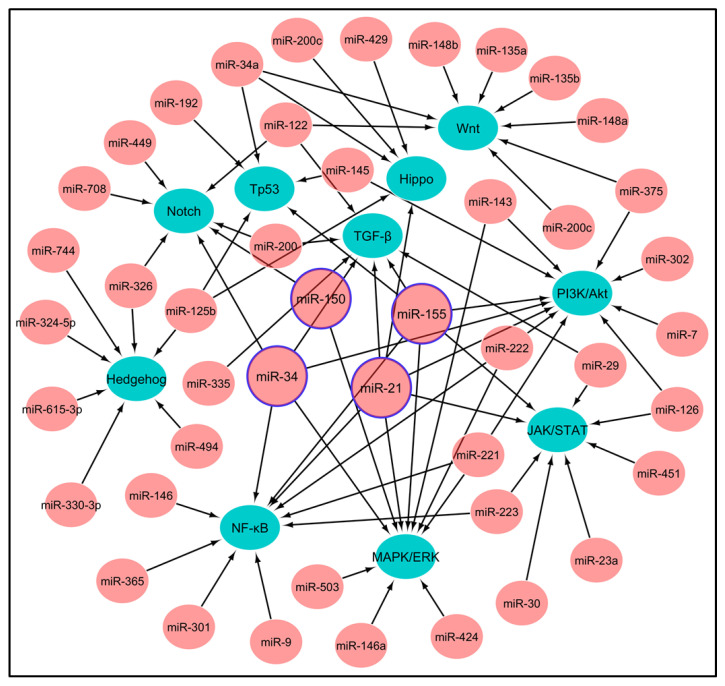
Illustrating network of the signaling mechanisms in lung cancer through miRNA-mediated regulation. The image shows the signaling pathways that are involved in lung cancer, including Wnt, TGF-β, Notch, Hedgehog, PI3K/Akt, MAPK/ERK, JAK/STAT, NF-κB, Hippo, and Tp53. Each of these signaling pathways is regulated by many miRNAs. Four specific miRNAs, namely miR-21, miR-150, miR-155, and miR-34, are known to have a significant impact on the regulation and progression of lung cancer, and they will be focused on by miRNA therapeutics.

**Figure 4 pharmaceutics-15-02061-f004:**
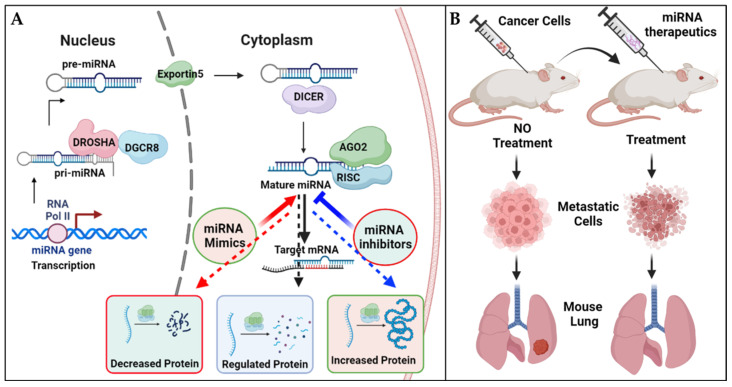
A detailed model illustrating the biogenesis of miRNAs and demonstrating the effectiveness of miRNA-based therapies for managing lung metastases. (**A**) The biogenesis of miRNAs involves transcription by RNA polymerase II, processing by Drosha and Dicer enzymes, and incorporation into the RNA-induced silencing complex (RISC) to regulate gene expression at both the cellular and animal levels through oncology-directed miRNA replacement therapy. (**B**) Experimental animals have been used to test the efficacy of miRNA-based treatments in restricting metastasis, with studies conducted to assess the ability of these therapies to prevent the spread of cancer to other parts of the body.

**Figure 5 pharmaceutics-15-02061-f005:**
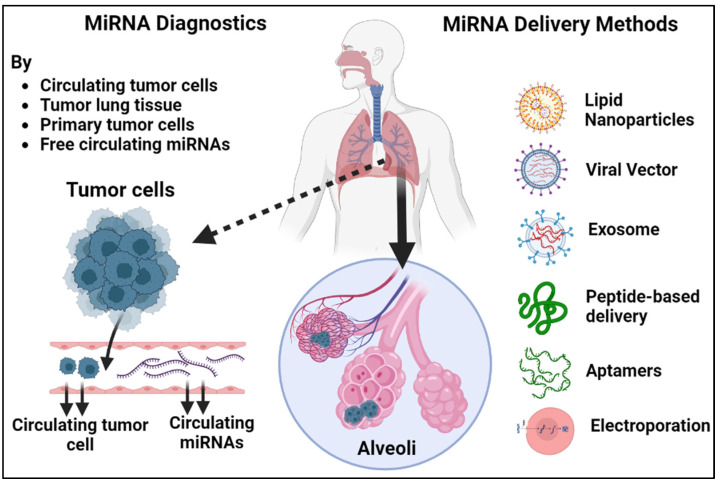
An illustrated guide to the diagnostic and therapeutic potential of miRNAs and methods for delivering miRNA therapeutics. MiRNAs can be extracted from circulating miRNAs, circulating tumor cells, primary tumor cells, and tumor lung tissue and analyzed for their expression patterns. These miRNA profiles can then be used to develop non-invasive diagnostic tools for cancer detection and monitoring and to guide personalized treatment strategies. The delivery methods include lipid-based nanoparticles, viral vectors, exosomes, aptamers, peptide-based delivery, and electroporation. Each method has its own advantages and limitations, and the choice of delivery method depends on factors such as the type of miRNA therapeutic and the target tissue.

## Data Availability

The data presented in this study are available in this article and Appendix A.

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
