# Peer review of "Unraveling Therapeutic Opportunities and the Diagnostic Potential of microRNAs for Human Lung Cancer"

_pharmaceutics, 2023, doi:10.3390/pharmaceutics15082061_

Round 1

Reviewer 1 Report

Thank you for the opportunity to review this interesting manuscript. 

Lung cancer is a major public health problem and this manuscript makes an important contribution to achieving the goal of early diagnosis.

Author Response

Dear reviwer,

 We would like to express our gratitude for reviewing our review article. Your time and effort in evaluating our work are sincerely appreciated. We value your expertise and the opportunity to have our article considered for publication. We have carefully reviewed the content and made necessary revisions to enhance its quality and clarity. We believe that these revisions have strengthened the overall manuscript. We remain open to any further feedback or suggestions you may have and look forward to the possibility of collaborating with you to ensure the excellence of our review article.

Reviewer 2 Report

The manuscript entitled “Unraveling Therapeutic Opportunities and Diagnostic Potential of microRNAs for Human Lung Cancer” focuses on the discussion of roles of microRNAs in diagnosis and therapy of lung cancers. Since lung cancer is one of the leader causes of cancer-related death over the world and there is a little progress in early diagnosis and chemotherapy options, discovery of novel biomarkers for lung cancer is of crucial importance. This manuscript summarizes the latest advancements in this field and could be of interest to the journal audience. However, there are some concerns, which have to be addressed before publication.

1.     The discussion of the roles and mechanisms of miRNAs in lung carcinogenesis in section 2 is illogical. MiRNAs can play both oncogenic and tumor-suppressor roles depending on a gene, which is under control of a definite miRNA. In subsection 2.1 and on Fig. 1B, do not use this term as a synonymous for oncogenic. Oncogenesis is a process resulted from mutations in genes, dysregulated angiogenesis, EMT, immunological control etc. The authors should separately discuss, which genes and processes are regulated by oncogenic miRNAs and which ones – by tumor suppressor miRNAs, not confusing them.

2.     The same is for sections 3 and 4 – confusing discussion of oncogenic and tumor-suppressive miRNAs and their effects (regulation of immune response and cell signaling).

3.     In subsection 2.2 and Figs 2 and 3, there are crucial mistakes regarding the interrelations between genes and miRNAs. That is miRNA controls a gene, but not vise versa. Therefore, in the text, it should be “miRNAs REGUATE GENES/PATHWAY”, but not “miRNAs regulated by genes/pathways” The same is for figures – the arrows should be from miRNAs to genes.

4.     There are many clumsy sentences such as on lines 68, 159-160, 249 (cellular communication in the body), 251 (small molecules of miRNAs), etc.

English requires editing

Author Response

Dear reviwer,

We would like to express our heartfelt thanks for the insightful feedback you provided on our review article. Your commitment to reviewing our work and the thoughtfulness of your comments are truly valued. With great appreciation, we have carefully considered each of your inquiries and made the appropriate revisions to address your concerns. Our responses, outlined below, demonstrate our efforts to effectively address the points you raised. We welcome any additional discussion or collaboration to enhance the accuracy and comprehensiveness of our review article.

Reviewer 3 Report

Fascinating and informative manuscript. The only comment/suggestion is to add one paragraph between section titles and subtitles. This paragraph should give a brief description of the current section. 

In the entire manuscript, I did not find any mention of artificial intelligence. Artificial intelligence can have a significant contribution to the diagnosis or classification of lung cancer type. Could you at least mention in one small paragraph some implementation of AI? 

Author Response

Dear reviwer,

We extend our sincere gratitude for your invaluable feedback on our review article. Your time and expertise in reviewing our work are greatly appreciated. Your thoughtful comments and suggestions have allowed us to improve the quality and clarity of our article. In our response below, we carefully address each of your inquiries and have incorporated the necessary revisions to address your concerns. We are confident that our responses adequately address the points you raised, and we are open to any further discussion or collaboration to ensure the accuracy and comprehensiveness of our review article.

Round 2

Reviewer 2 Report

The authors have made all corrections in the manuscript

No comments